



# Comparison Metrics Microscale Simulation Challenge for Wind Resource Assessment

Florian Hammer[1], Sarah Barber[1], Sebastian Remmler[2], Federico Bernardoni[3], Kartik Venkatraman[4], Gustavo A. Díez Sánchez[6], Alain Schubiger[1], Trond-Ola Hågbo[4,5], Sophia Buckingham[4], and Knut Erik Giljarhus[5]

[1]Eastern Switzerland University of Applied Sciences, Oberseestrasse 10, 8640 Rapperswil, Switzerland
[2]Wobben Research and Development GmbH, Borsigstr. 26, 26607 Aurich, Germany
[3]UTD Wind, University of Texas at Dallas, 75080 Richardson, Texas, USA
[4]von Karman Institute for Fluid Dynamics, 1640 Sint-Genesius-Rode, Belgium
[5]University of Stavanger, 4021 Stavanger, Norway
[6]Independent researcher

**Correspondence:** Florian Hammer (florian.hammer@ost.ch)

**Abstract.** The main goals of a wind resource assessment (WRA) at a given site are to estimate the wind speed and annual energy production (AEP) of the planned wind turbines. Several steps are involved in going from initial wind speed estimations of specific locations to a comprehensive full-scale AEP assessment. These steps differ significantly between the chosen tool and the individuals performing the examination. The goal of this work is to compare different WRA simulation tools at the Perdigão site in Portugal, for which a large amount of wind measurement data is available, in terms of both accuracy and costs. Results from nine different simulations from five different modellers were obtained via the "IEA Wind Task 31 Comparison metrics simulation challenge for wind resource assessment in complex terrain", consisting of a range of linear models, Reynolds-Averaged Navier-Stokes (RANS) computational fluid dynamics models and Large Eddy Simulations (LES). The wind speed and AEP prediction errors for three different met mast positions across the site were investigated and further translated into relative "skill" and "cost" scores, using a method previously developed by the authors. This allowed the most optimal simulation tool in terms of accuracy and cost to be chosen for this site. It was found that the RANS simulations achieved very high prediction accuracy at relatively low costs for both wind speed and AEP estimations. The LES simulations achieved great wind speed prediction for certain conditions, but at a much higher cost, which in turn also reduced the number of possible simulations, leading to a decrease in AEP prediction accuracy. For some of the simulations, the forest canopy was explicitly modelled, which was proven to be beneficial for wind speed predictions at lower heights above the ground, but lead to under-estimations of wind speeds at upper heights, decreasing the AEP prediction accuracy. Lastly, low correlation qualities between wind speed and AEP prediction error were found for each position, showing that accurate wind modelling is not necessarily the only important variable in the WRA process, and that all the steps must be considered.



# 1 Introduction

## 1.1 Wind Resource Assessment tools

The main goals of a wind resource assessment (WRA) at a given site are to estimate the wind speed and annual energy production (AEP) of the planned wind turbines. Estimations of AEP over the wind farm's life expectancy (of usually 20-25 years) determine the expected rate of return, the main driver for investors and wind farm owners (FGW, 2007). Wind speed estimations commonly serve as the first step in estimating the AEP. There are several steps involved before arriving at a final AEP estimation, with its corresponding uncertainty, that can differ between simulation tools and between wind resource assessors. This includes wind measurements, wind data processing, correction and analysis, long-term wind resource extrapolation, wind resource vertical extrapolation, wind resource horizontal extrapolation, energy production calculation, estimation of the wake effects and estimation of the losses (Barber et al., 2022a). Each of these steps comes with its own uncertainties, the understanding of which is paramount for estimating the project risks.

The task of wind speed estimation and the whole process of AEP estimation becomes even more challenging at complex sites (Clifton et al., 2022; Bowen and Mortensen, 1996; Wood, 1995; Pozo et al., 2017). This is due to the complex effects occurring for flow over irregular and steep surfaces such as mountainous terrain. These effects may include an increase in turbulence, sudden changes in wind speed and direction, flow separation, recirculation regions and more - and these can all negatively affect the performance of wind turbines. Some of these complex phenomena can even be beneficial in terms of power production, such as katabatic winds (Parish and Cassano, 2003).

As a result of these various complexities, a vast range of AEP estimation tools are available, serving different needs with regard to accuracy and costs. AEP estimation tools range from simple linear models, as used in WAsP (Troen and Lundtang Petersen, 1989), for example, that are easily and readily set up and require very little interaction with the user and directly report the AEP, to Large Eddy Simulations (LES) (Beaucage et al., 2014; Barber et al., 2022a), requiring a thorough understanding of the used models and access to high-performance computing facilities. Simulation tools also differ in spatial and time resolutions, ranging from directly simulating or modelling microscales to focusing on the mesoscale wind patterns. More elaborate tools such as LES allow for very specific set-ups, for example for flow from certain wind directions over short time periods. Such an approach might lead to very accurate wind speed estimations, but leaves the user with a variety of options and uncertainties for the AEP estimation. Furthermore, such simulations are computationally prohibitive for many users.

Hence, it is important to provide an overview of available simulation tools and to quantify the accuracy and uncertainty for a given site, as well as the associated costs.

## 1.2 Evaluation and comparison of tools

There have been various undertakings for the evaluation of modelling tools, which can be split up into two main groups: 1) Evaluation of Computational Fluid Dynamics (CFD) tools for wind modelling (Barber et al., 2020b; Britter and Baklanov, 2007; AIAA, 1998; Daish et al., 2000; VDI, 2005; Bechmann et al., 2011; Berg et al., 2011; Bao et al., 2018; Menke et al., 2019); 2) Evaluation of workflows for AEP estimation (Lee and Fields, 2021; Barber et al., 2020a; Mortensen et al., 2015;





Rodrigo et al., 2018). The first group is valuable in terms of flow estimation and optimisation of flow models, however, is not commonly concerned with WRA, as this requires several more steps. On top of this, accurate flow estimation need not necessarily result in accurate estimations of the wind resource itself, as recently demonstrated by the present authors (Barber et al., 2022a). The second group addresses the important topic of WRA, however is often plagued by poor information disclosure due to confidentiality issues and therefore only of limited benefit. In order to circumvent these drawbacks, the present authors recently conducted a study where six different simulation models, including one LES model, for the Bolund Hill site were compared in terms of both wind speed and AEP estimation accuracy and costs Barber et al. (2020c). The comparison was done based on accuracy and cost scores that allow users to compare different simulation tools and to choose the most suitable tool according to the available budget and the required accuracy (Barber et al., 2020a). This methodology was further developed in a more recent study and extends to the whole decision process for choosing a WRA workflow (Barber et al., 2022a). In this study seven different WRA workflows for five different complex terrain sites were compared in terms of both wind speed and AEP estimation accuracy and costs. The authors showed that there is no linear relationship between the wind profile prediction accuracy and the AEP prediction accuracy due to the large differences in the WRA workflows as well as due to the varying importance of wind speed on AEP depending on its frequency and magnitude within a particular wind speed sector.

### 1.3 Goal of this work

The goal of this work is to further enhance the understanding and building of a knowledge base for choosing the most optimal WRA tool for a particular complex terrain site based on accuracy and costs. For this we chose one site with an inhomogeneous surface and complex wind conditions, which was simulated by five different simulation tools as part of a simulation challenge within IEA Wind Task 31 (Barber et al., 2020a), ranging from linear models to LES. As the main focus of the previous work was the simulation and comparison of many sites, but restricted in the amount of used tools, this work's objective is the comparison of a number of tools for one specific site.

In section 2 of this paper we present the complex site for the simulations, how the comparison between the tools was done and show an overview of the simulation tools in this study. The results are discussed in section 3, where we also try to provide the reasons for the differences in performance and how they might be mitigated. Differences within given simulation tools with various models and set-ups applied are discussed as well. Lastly, some conclusions are drawn in section 4.

### 2 Methodology

In the following sections the chosen site for the wind speed profile and AEP comparisons is presented. The calibration and simulation set-ups are crucial for a successful comparison, and they are therefore discussed as well. This is followed by the comparison methodology based on the skill and cost scores introduced by Barber et al. (2020c). The methodology section is concluded with an overview of all simulation tools for the wind speed and AEP estimations. The simulations for this work were conducted by various modellers as part of the "IEA Wind Task 31 Comparison metrics simulation challenge for wind resource





assessment in complex terrain" (Barber et al., 2020a). It is an important point to note that some tools only estimate the wind
85 speed, and the AEP values have to be calculated by the modellers in additional steps. These steps are given for each tool where
this applies.

## 2.1 The site

The Perdigão site in Portugal (Fernando et al., 2019) was chosen due to the volume and quality of available measurement
data, the complexity of the terrain and the relative lack of simulations already carried out. A large measurement campaign was
90 undertaken between 15 December 2016 to 15 June 2017 as part of a large EU-US collaborative field experiment (Fernando
et al., 2019). Measurement data from many met masts as well as remote sensing devices, ground sensors and noise monitoring
sensors is available (Gomes et al., 2017). The site consists of flow over two parallel ridges with SE-NW orientation, which are
4 km long and 500-550 m tall and separated by about 1.5 km. The two main wind directions are approximately perpendicular
to the ridges. A 3D representation of the site as well as an overview of the met masts with heights of 60 m (orange) and 100 m
(red) are shown in Fig. 1. A 2 MW ENERCON E-82 wind turbine is located close to met mast 20, marked with "WTG".

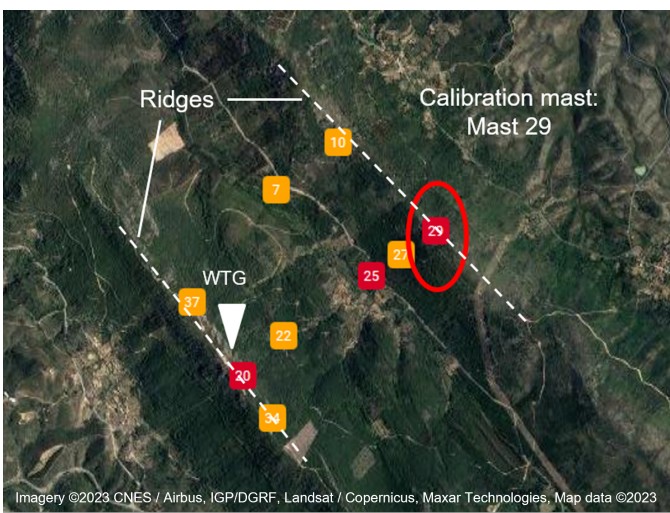

**Figure 1.** The Perdigão site in Portugal. Two parallel ridges with SE-NW orientation, which are 4 km long and 500-550 m tall and separated
by about 1.5 km. Depicted are nine met mast locations with a height of 60 m (orange) and 100 m (red).


In order to demonstrate the complexity of the flow patterns, wind roses for each met mast at the top positions are depicted in
Fig. 2 over the entire measurement period of each mast, plotted using the data available in Gomes et al. (2017). It is expected
to be fairly difficult to estimate the flow correctly for all the simulation tools.

For the simulation tool comparison within this work we chose the three met masts with height 100 m, as shown in Fig. 2.
Data for the period from UTC 9 March 2017 01:10:00 to UTC 17 June 2017 18:10:00 was available. The three masts, denoted
by the numbers 29, 25 and 20, are positioned on a straight line along the main wind direction. Met masts 29 and 20 show very
similar wind roses, whereas met mast 25 is dramatically different, having much lower wind speeds, which is expected due to



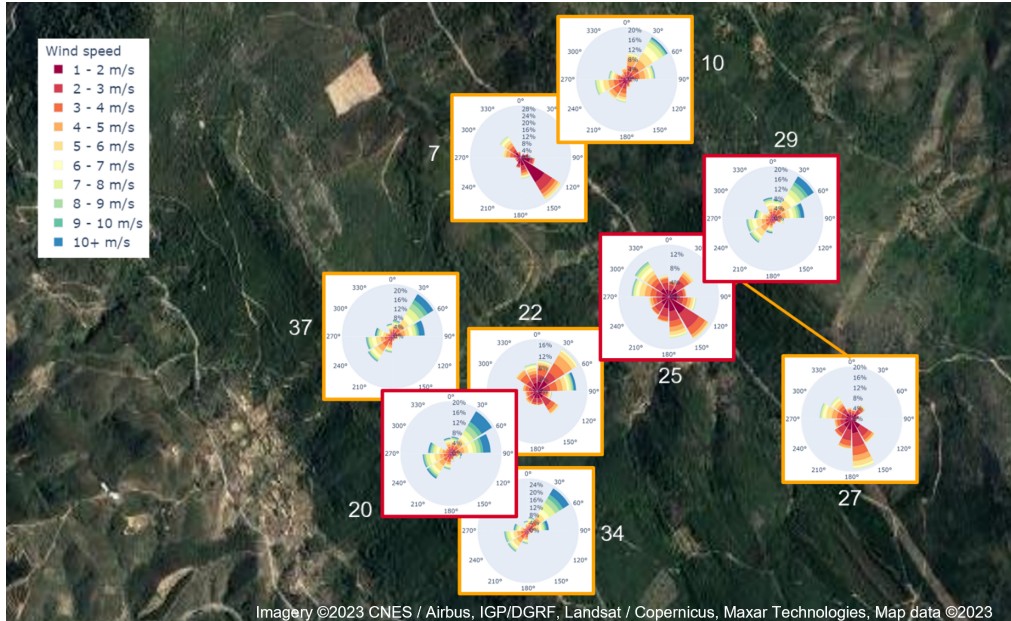

**Figure 2.** Nine wind roses for the different met mast locations. The wind roses are shown for their respective top position.

its location between the two ridges. An interesting point to note is that the main wind direction 800 m above the ridge is the SW direction, but due to a mesoscale circulation the flow closer to the surface comes from the NE direction at certain times

of day, which is what we see in the wind rose of met masts 29 and 20. Additionally, the presence of the valley forces wind to travel in a SSE direction, reflected in the wind rose for mast 25.

The mean measured wind profiles for these three masts over the entire measurement period are shown in Fig. 3, plotted using the data available in Gomes et al. (2017), where the points denote the vertical positions of the anemometers and the lines a logarithmic fit (Barber et al., 2020a).

In order to set up the correct initial and boundary conditions for the CFD simulations a 'calibration mast' was determined. Details of the calibration process are given in Section 2.3 for each simulation workflow. The simulation accuracy can then be assessed by comparing the simulation results to measurements at a different location ('validation mast'), which is ideally far away from the calibration mast to allow for the flow to develop through the simulation domain. In order to reduce calibration inaccuracies, one of the met masts on the ridge was chosen with the most frequent wind speeds coming from a direction over

a more homogeneous terrain. This ensures that for the calibration process no simulation model is affected by complex terrain and other obstacles. Hence, the choice for the 'calibration mast' was made for met mast 29. The estimated wind speeds from the various simulation tools were matched with the wind speed measured at a height of 80 m at met mast 29.

Met masts 25 and 20 serve as the 'validation masts' in order to assess the capabilities of the different tools. Met mast 25 tests the capabilities in terms of flow over separated regions with strong flow turning and wind speed changes, whereas met mast 20

reveals strengths and weaknesses for simulating flow on top of hills. These two met masts are also in line with the main wind



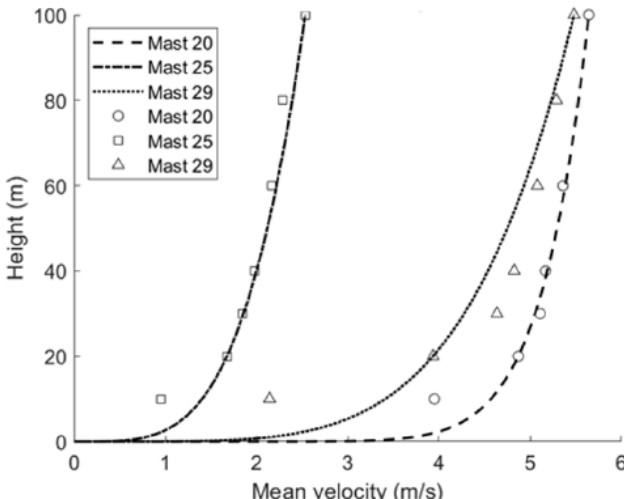

**Figure 3.** The mean measured wind profiles for met masts 20, 25 and 29 over the entire measurement period. The points denote the vertical positions of the anemometers and the lines a logarithmic fit (Barber et al., 2020a).

direction measured at met mast 29. Additionally, met mast 20 is close to an installed WTG on the site and hence of interest in terms of wind profile and AEP estimations.

## 2.2  Skill and cost scores

As mentioned above, the skill and cost scores developed by Barber et al. (2022b) are used in this study in order to compare
the simulation tools in terms of wind speed and AEP accuracy and costs. The cost scores of the simulations are determined based on a questionnaire, taking into account the required number of simulations, the required computational time, the cost of a simulation per hour and per core as well as the set-up and post-processing time spent. This is described by the following equation.

$$s_{\text{tot}} = l + t + (s_{\text{setup}} + s_{\text{run}} + s_{\text{post}}) \cdot n_{\text{sim}} \tag{1}$$

with the software licensing costs per project, $l$, the staff training cost per project, $t$, the simulation setup cost, $s_{\text{setup}}$, the simulation run-time cost, $s_{\text{run}}$, the post-processing cost, $s_{\text{post}}$, and the number of simulations, $n_{sim}$. As for some of the workflows an additional step for the AEP predictions are needed, the cost of this effort is added to $s_{\text{tot}}$. The staff training cost per project, $t$, the simulation set-up cost, $s_{\text{setup}}$ and the post-processing cost, $s_{\text{post}}$, all depend on the hourly rate of the modeller and the time spend for each task.



The skill scores are divided into two, namely the skill score based on the wind speed estimations, $S_U$, and based on the AEP estimations, $S_E$. The score based on wind speed is defined by

$$S_U(U_{\mathrm{rmse}}) = \begin{cases} \frac{3 - U_{\mathrm{rmse}}}{3} & \text{if } U_{\mathrm{rmse}} \leq 3 \\ 0 & \text{if } U_{\mathrm{rmse}} > 3 \end{cases} \qquad (2)$$

with

$$U_{\mathrm{rmse}} = \sqrt{\sum_{i=1}^{N} \frac{(U_{s,i} - U_{\mathrm{meas},i})^2}{N}}. \qquad (3)$$

Here, $U_{s,i}$ is the estimated wind speed at anemometer position $i$ of a chosen met mast, whose wind speed at the same position is defined by $U_{\mathrm{meas},i}$. The deviation error value of 3 was chosen because it corresponds to the maximum observed wind speed error between the simulations and the measurements. For each comparison study a sensible value should be selected and kept constant throughout for meaningful results. However, the absolute value is not important because only relative errors between different tools are used.

The score based on AEP is defined by

$$S_E(E_s, E_{\mathrm{meas}}) = 1 - \left| \frac{E_{\mathrm{meas}} - E_s}{E_{\mathrm{meas}}} \right| \qquad (4)$$

with the estimated AEP value $E_s$ at a chosen met mast position, whose AEP based on the measurements is given by $E_{\mathrm{meas}}$. The AEP estimation is based on a power curve of the 2 MW ENERCON E-82 wind turbine. In order to calculate the expected AEP of this wind turbine at a given validation location, the time series of ten-minute averaged wind speeds was multiplied

by the power curve for the entire measurement period. For the measurement values $U_{\mathrm{meas},i}$ and $E_s$ for the calibration and validation met masts a data period from UTC 9 March 2017 01:10:00 to UTC 17 June 2017 18:10:00 was chosen in order to ensure overlapping time periods between all the masts. Further details about the skill and costs score can be found in Barber et al. (2020c, a, 2022a, b).

### 2.3   Simulation tools and workflows

A variety of simulation tools and workflows were used for the purpose of this comparison, ranging from simple linear models run on a normal computer to LES requiring high performance computing in order to be of value. This range of simulation tools and workflows is expected to also yield a range of cost and skill scores. The biggest challenge for conducting a study like this is the simulation set-up and the provision of the required data and information for doing so. In order to successfully set up different kinds of simulations that are comparable a number of points needed to be considered and provided to each modeller:

– Topography and roughness maps.

– Description and set-up of measurement equipment.

– A power curve for the AEP estimation.



– A clear definition about the method of calibration.

An overview and details of the simulation tools and workflows used for comparison are shown in Table 1. In total five
different simulation tools from six different modellers were used with various configurations and conditions resulting in a total
of nine different simulation set-ups. In the following sections the various simulations are denoted by the name given in the
respective table row. The used simulation tools, the simulation set-ups and the workflows for the AEP estimation are explained

**Table 1.** In total five different simulation tools from six different organisations were used with various configurations and conditions resulting
in a total of nine different simulation set-ups

| Model / Wind tool | PCE-LES | OpenFOAM | E-Wind | ANSYS Fluent | WAsP |
|---|---|---|---|---|---|
| **Organisation** | University of Texas at Dallas | Von Karman Institute | Enercon | OST | Independent researcher |
| **Reference names** | UTD_PCE_LES  UTD_PCE-canopy_LES | VKI_OpenFOAM_k-$\epsilon$-structured  VKI_OpenFOAM_k-$\epsilon$-unstructured | ENERCON_E-Wind_k-$\epsilon$  ENERCON_E-Wind_k-L  ENERCON_E-Wind_k-$\omega$ | ost_fluent_k-$\omega$-SST | org04_wasp_sim01 |
| **Flow model** | LES | RANS | RANS | RANS | Linearized Flow Model |
| **Turbulence model** | Smagorinsky (Smagorinsky, 1963) | k-$\epsilon$ | k-$\epsilon$    k-L    k-$\omega$ | SST k-$\omega$ | - |
| **Number of sectors** | 8 | 12 | 12 | 12 | 12 |
| **Common application range** | Non-flat terrain | Non-flat terrain | Non-flat terrain | Non-flat terrain | Only for flat terrain |
| **Relative simulation time** | High | Medium | Medium | Medium | Low |
| **Relative set-up complexity** | High | Medium | Medium | Medium | Low |

in the following.





- **PCE-LES**:

PCE-LES is a method that aims to predict the turbine power production and the wind velocity at any given location of interest using a Polynomial Chaos Expansion (PCE) surrogate model based on Large Eddy Simulations (LES). The PCE is used to fit the data of the numerical simulations into a surrogate model able to predict each quantity of interest (e.g. wind velocity at a given location) as functions of the wind speed ($U_{29}$) and wind direction ($\theta_{29}$) measured at the met mast 29. The PCE consists of a multivariate orthonormal polynomial bases (Xiu and Karniadakis, 2002) over which the given random process is projected.

For this study, eight different wind directions and eight different wind velocities were simulated in order to reproduce the variability at the site. Each simulation was performed with our in-house code able to perform LES of wind turbines and terrain topography by solving the filtered non-dimensional governing equations for incompressible flow. The numerical discretisation, described in Orlandi (2000), consists of a staggered central second-order finite-difference approximation in a Cartesian coordinate system cumulative with a hybrid low-storage third-order Runge-Kutta scheme to advance the equations in time. The forces of the rotor acting on the flow are reproduced using the rotating actuator disk model (Ciri et al., 2017, 2018). The terrain topography, the towers and the nacelles are reproduced using the Immersed Boundary Method (IBM) implemented by Orlandi and Leonardi (2006) and by Santoni et al. (2015, 2017). Each simulation reproduces a region centered on the met mast 29 of about 5 km by 4 km by 2 km respectively in the streamwise, spanwise and vertical direction. The resolution in the horizontal directions is about 4m while in the vertical direction a resolution of 2.5 m was adopted close to the topography and the turbine. Two different topography models were used. The first model reproduces only the terrain topography while the second integrates also the vegetation canopy. Given the high computational cost of reproducing the actual vegetation, the canopy is simulated with a distribution of stems on top of terrain topography. The stem size and distribution reproduce the same canopy frontal solidity (Monti et al., 2019; Nepf, 2012) of the actual vegetation at the site extracted from the Corine Land Cover (European Union, 2018) database. The PCE surrogate model based on high fidelity simulations is then used to reproduce the stochastic variability of the wind conditions at the met mast 29 in order to obtain the average wind velocity profile and the turbine power production in each wind sector at each location of interest.

- **OpenFOAM**:

All simulations were performed using the OpenFOAM v2012 flow solver. The steady-state incompressible flow solver simpleFoam was utilised, and no thermal effects were considered.

For this site, a $7.5 \text{ km} \times 7.5 \text{ km}$ terrain patch was simulated around the calibration met mast 29 using the terrain map from the Shuttle Radar Topography Mission (SRTM) database.

A structured grid was generated using the toolbox `terrainBlockMesher` developed by Schmidt et al. (2012). Additionally, an unstructured grid was generated using the tool `snappyHexMesh` for comparison purposes. The structured mesh comprises of 12.7 million cells, which pertains to 227 x 227 x 120 cells across the terrain patch. The height of the





first cell close to the ground is close to 3 m. The vertical mesh resolution is set to 33 m, with a uniform stretching applied to increase the mesh resolution close to the terrain.

The unstructured mesh has 17.8 million cells. It is refined incrementally toward the terrain and has an overall stretching in the vertical direction. The height of the cells adjacent to the ground is, like the structured grid, close to 3 m.

For all the simulations, the effect of forest canopy was modelled using source terms in the momentum equation. They were modelled as a porous medium, as detailed in Costa (2007), by selecting a mean tree height of 3 m. A cell set was utilized in order to select a volume of cells 3 m above the ground for the whole domain, which then forms a canopy zone where the source terms were activated. The following model parameters were used: Porosity surface area per unit

volume or the leaf area density $\sigma = 1.0$, drag coefficient ($C_d = 0.25$), and the Power law model exponent coefficient ($C1 = 2.0$). Simulations were performed for 12 different sectors and calibrated to match wind speed and direction at a height of 100 m at met mast 29. Simulation profiles were then extracted for the comparison with the measurements. In order to calculate the AEP the wind speeds for each sector at height 80 m were determined and then processed using the open-source library WindPowerLib (Haas et al., 2019). The calculations were performed for three months worth of data

and then scaled to one year to yield the AEP.

– **WAsP**:

WAsP is one of the most common simulation tools used in the wind industry to estimate wind resources. Its model is based on linearised equations for neutral conditions, although atmospheric stability can be taken into account (Magdalena et al., 2016). For this simulation, a contour map was generated from a *.tiff file, showing the elevation contours with a

horizontal resolution of 20 m. Furthermore, a *.map file was obtained with a domain size of 5 km by 5 km, centred around met mast 29. The roughness was generated through a complete roughness file from the Corine database; the individual roughness lengths were attributed to each land (Silva et al., 2007) and optimised with satellite images from Google Earth. The direction variable was obtained with respect to the true north from the 'U´ and 'V´ wind speed variables and converted from radians to degrees (NCA, (accessed July 1, 2020). To validate this method, a comparison was made with

data and wind roses of long-term reanalysis datasets, such as MERRA2 and ERA-5. The coefficient of determination was R2 > 0.8 and the wind roses were very similar. Once the data was obtained and properly formatted, some basic filtering was performed for periods with constant line values for some variables. Additionally, the atmospheric stability was checked by means of the daily scale, Pasquill-Gifford Stability Class correlated with Monin-Obukov length, and showed mostly stable conditions (Woodward, 1998). An adjustment was made due to the higher values of RIX for the complex

site (Mortensen et al., 2006) This particular adjustment was used to optimize the wind speed and energy estimation based on the power curve of the ENERCON E-82 turbine. Previously, a long-term correction factor was applied with ERA-5 reanalysis datasets.

– **E-Wind**:



E-Wind is the ENERCON in-house CFD solution for RANS flow modelling in site assessment. It combines all necessary steps for extrapolating a time series measured on site to the planned turbine positions in a highly automated way. This includes grid generation, simulation, shear calibration and prediction of free wind speed time series as well as AEP calculation. For more details on the methodology, see Alletto et al. (2018). E-Wind uses a single circular structured grid for all wind directions. The grid is generated using the Pointwise grid generator. It has a constant horizontal cell size of 25 m in the area of the wind farm and stretched cells towards the outer domain boundaries. The grid is fitted to the terrain with a vertical cell size of 1m at the surface and vertical stretching towards the upper boundary. The domain size is chosen such that every point of interest has a distance of at least 10 km to the boundaries. In total, we used 1.8 million cells for the simulations presented here. E-Wind employs the open source toolbox OpenFOAM to solve the steady- state 3D RANS equations. The solver includes the effect of forests, Coriolis force, and buoyancy in the turbulence equations. The thermal stability is modelled by a surface heat flux to account for stable (negative heat flux) or unstable (positive heat flux) atmospheric conditions. The flow is driven by a prescribed geostrophic wind speed and wind direction at the upper domain boundary. By default, we use 24 geostrophic wind directions. For each geostrophic wind direction, the simulated shear at the reference mast is automatically compared to the measured wind shear for the respective local wind direction and the surface heat flux is automatically adjusted until a sufficient matching between measured and simulated shear is obtained. For the prediction of time series at the turbine positions, E-Wind creates a look-up table of speed-up factors and wind-direction deviations between the reference point and the respective target positions, depending on the local wind direction at the reference position. For each time stamp, we multiply the measured wind speed by the simulated speed-up factor and add the wind-direction deviation to the local wind direction in order to predict wind speed and wind direction at each target position. For the prediction of AEP, we use the time series of predicted wind speed at the turbine hub position and multiply it with the turbine power curve. In cases where the measured time does not cover a full year, we scale the predicted energy production accordingly.

– **Fluent RANS**:

The commercial CFD software ANSYS Fluent allows for simulations of various different industrial problems, such as combustion applications, fluid structure interaction, multiphase flows and more, by either RANS, LES or DES simulations. Within this work, RANS simulations with the SST k-$\omega$ turbulence model (Menter, 2012) were carried out. 12 different wind directions were simulated, for each of which the wind speed profiles and the turbulent kinetic energy as well as the turbulent dissipation rate were determined through the roughness height, as described in Richards and Hoxey (1993). In order to reduce the setup and post-processing times, and hence, the associated costs, a simulation workflow, elaborated in detail in Barber et al. (2020b), was used. A 5 km by 5 km square region with a height of 1.4 km around met mast 29 was chosen for the computational domain. For the inflow condition a logarithmic wind speed profile based on an assumed roughness height of $z_0 = 0.75$ m was created. A wind speed of 10 m/s at a height of 500 m above ground was set. By means of a grid independence study for wind direction 270°, with cell sizes ranging from 7.5 to 25 m, the optimal resolution of 15 m was determined, resulting in a total cell count of 2.5 million cells. After carrying out all simulations,





the predicted wind speeds in each sector were scaled to match the measured wind speeds at height 80 m of met mast 29. In order to calculate the AEP values, firstly, the flow turning and speed-up factors between the *predicted* values at the validation masts and the *predicted* values at the calibration mast for each sector were determined. After that the *measured* ten-minute averaged wind speeds at the calibration mast 29 were multiplied by the speed-up factors for the validation masts. Based on the new time series for the validation locations and the given power curve of the ENERCON E-82 wind turbine, the energy production was calculated and scaled to account for a period of one year.

## 3 Results and discussion

The wind speed and AEP estimation outputs resulting from the different simulation tools were standardised by means of a results template provided to the modellers. The results template serves as an interface for the evaluation and comparison process and reduces the risk of errors. For each simulation run, the various modellers stored 3D wind vector components of vertical wind speed profiles at the calibration and validation met mast locations for each 30° wind direction sector as well as the calculated AEP for each sector. In this section, the wind speeds are first examined, and then the AEP results. Finally, the relation between the wind speed and AEP prediction errors are investigated.

### 3.1 Wind speed

#### 3.1.1 Root-mean-square errors

In the following the root-mean-square errors (RMSE) between the measured and the simulated wind profiles are presented, which are used for the calculation of the skill score $S_U$.

Fig. 4, Fig. 5 and Fig. 6 show RMSE values between the simulated and the measured wind profiles for different evaluation heights for met mast 29, met mast 25 and met mast 20, respectively, for each simulation set-up. In case of a single evaluation point, e.g. at 80 m, this expression simply reduces to the absolute difference. The dashed lines represent the mean RMSE values for all simulations at the specific height profile. For each of the twelve sectors the RMSE values were calculated for each simulation. The bars represent the mean value of the twelve sectors whereas the vertical bars represent the standard deviation.

The first thing to notice is that the lowest errors occur at met mast 29, which is not surprising as it serves as the calibration location. For the evaluation height 10 - 100 m, considering points closest to the ground, average RMSE values of $0.8 \, m/s$ can be observed. A decrease in RMSE of almost 60% is apparent for evaluation height 40 - 100 m. This significant difference is due to the fact that the simulated and the measured wind profiles have larger deviations in the region close to the ground and hence increase the RMSE.

Here, for example, a special roughness model seems beneficial by comparing UTD_PCE_LES versus UTD_PCE-canopy_LES, where a canopy model was used in the latter case and improved the accuracy of the wind speed profile prediction by around 25%. For the OpenFOAM simulations (VKI_OpenFOAM_k-$\epsilon$-structured, VKI_OpenFOAM_k-$\epsilon$-unstructured) a canopy model





was used as well, showing the lowest overall errors for the 10 - 100 m evaluation height. Interestingly, VKI_OpenFOAM_k-
$\epsilon$-structured with the structured grid performs slightly better than VKI_OpenFOAM_k-$\epsilon$-unstructured. This could possibly be
due to the forest canopy effects being more uniformly applied across the entire domain in case of the structured grid.

The lowest error values for all simulations are achieved at the single evaluation point at 80 m, resulting in a further reduction
of 30% compared to evaluation height 40 - 100 m.

At met mast 25 the overall RMSE values for the 10 - 100 m evaluation height are almost 15% higher than for met mast 29.
For the single evaluation point, at 80 m, the RMSE values increase by up to 200%. These considerable differences are due to the
extreme flow turning as visible from the wind roses and hence the increased difficulty to simulate the wind profiles accurately.

For UTD_PCE_LES and UTD_PCE-canopy_LES an even higher increase of 300% can be observed. This is mainly due to
two main factors. Firstly, only eight instead of twelve LES simulations were carried out, serving as deterministic samples to
build the surrogate model that captures the entire wind rose. The second major reason is the absence of the atmospheric bound-
ary layer (ABL) stratification as an input parameter for the surrogate model. Boundary conditions for the ABL stratification far
from the available met mast location are difficult to prescribe without using nested mesoscale models, which allow to capture
the influence of the topography over a larger region.

For met mast 25, the different evaluation heights have less impact on the error values. The simulations done with E-Wind
and WAsP (org04_wasp_sim01) show the lowest errors.

For met mast 20, the simulation results with OpenFOAM (VKI_OpenFOAM_k-$\epsilon$-structured, VKI_OpenFOAM_k-$\epsilon$-unstructured)
show errors of roughly 100% higher than for met mast 25. Interestingly, simulations with E-Wind and the different tur-
bulence models (ENERCON_E-Wind_k-$\epsilon$, ENERCON_E-Wind_k-L, ENERCON_E-Wind_k-$\omega$) result either in a 20 - 30%
improvement or reduction depending on the used model compared to met mast 25. The most surprising candidate, however,
is org04_wasp_sim01, showing great performance for being the most simple model and also being commonly considered
unsuitable for terrains such as the one seen at the Perdigão site (Bowen and Mortensen, 1996).

### 3.1.2 Skill and cost scores

Fig. 7 shows the relative skill versus relative cost scores based on wind speed for met mast 29, 25 and 20 at evaluation height
10 - 100 m. For the other evaluation heights a similar trend was observed and hence is not depicted for the sake of brevity. Here
"relative" means that all skill scores were normalised by the highest achieved score at the respective met mast. As the depicted
scores are only valid for the specific site under consideration, the relative score values rather than absolute values were chosen
in order to evaluate the performance differences of the different simulation tools. The light and darker red regions indicate
possible ranges of insufficient or unacceptable scores, respectively, that can be defined by the modeller.

Here, for example, relative skill scores below 40% would not be worth considering no matter how low the cost. In turn relative
cost scores above 60% would be deemed too costly independent of the simulation accuracy. These thresholds are arbitrary
and can be adjusted by a modeller depending on the available budget, computational power, time and required accuracy. A
successful WRA project could be used as a reference in this case, setting a relative cost score of 20% and a relative skill score
of 70%. From there it can be assessed if a doubling in cost would justify an increase in accuracy by 10% points, for example.

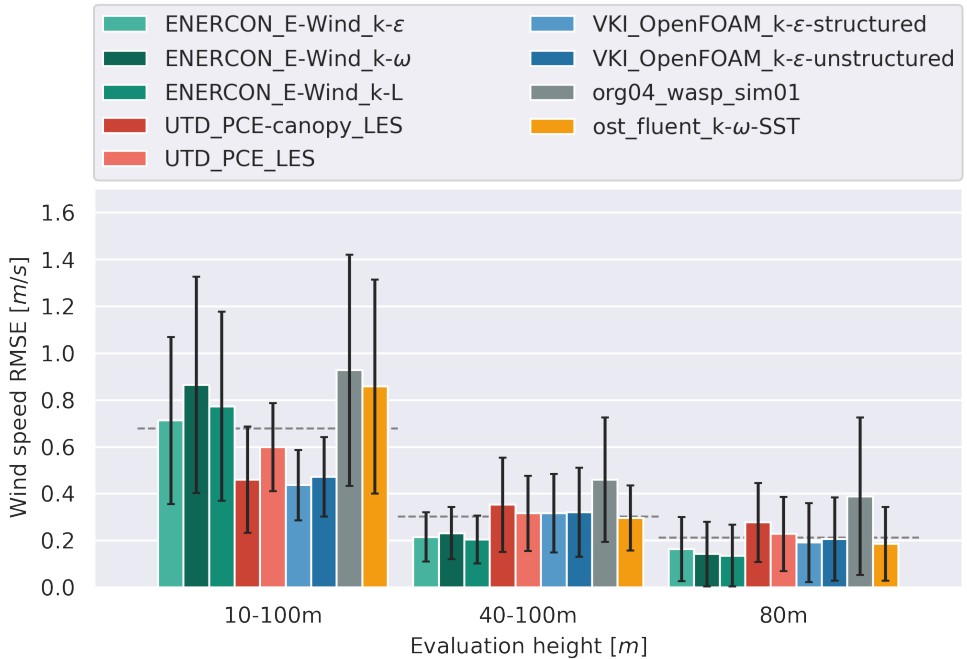

**Figure 4.** Root-mean-square errors for different wind profile evaluation heights for met mast 29. The colored bars denote the average RMSE of the wind profiles in each of the twelve sectors. The vertical black lines denote the standard deviation of the error among the twelve sectors. For evaluation height 80 m the RMSE reduces to the difference between the predicted and measured wind speed.

For met mast 29 the highest before skill scores, based on the answer of the questionnaire before running the actual simulations, are achieved by both LES simulations. This is followed by the three E-Wind RANS simulations. Whereas the scores for the RANS simulations VKI_OpenFOAM_k-$\epsilon$-structured, VKI_OpenFOAM_k-$\epsilon$-utructured and ost_fluent_k-$\omega$-SST are considerably lower. The main reason for this difference are the different experience levels of the modellers. Also the costs, especially in case of ost_fluent_k-$\omega$-SST, are significantly higher. The reason for this are the higher licensing costs of the used software and the lower level of automatisation, which increases the time spend by the modeller setting up and post-processing the simulations. The lowest before skill and cost scores are estimated for the WAsP simulation, being the simplest model and requiring less time for setting up and running the simulation. As LES simulations need a lot of computational power and time regarding set-up and processing, the UTD_PCE_LES and UTD_PCE-canopy_LES have the highest before cost scores. As the before skill and cost score evaluation is done for the whole site and not specific locations, the scores are the same for all three met masts.

As to the after skill and cost scores, a good agreement for the E-Wind simulations can be observed for all three met masts. For met masts 20 and 25 the UTD_PCE_LES and UTD_PCE-canopy_LES before skill scores were considerably overestimated,

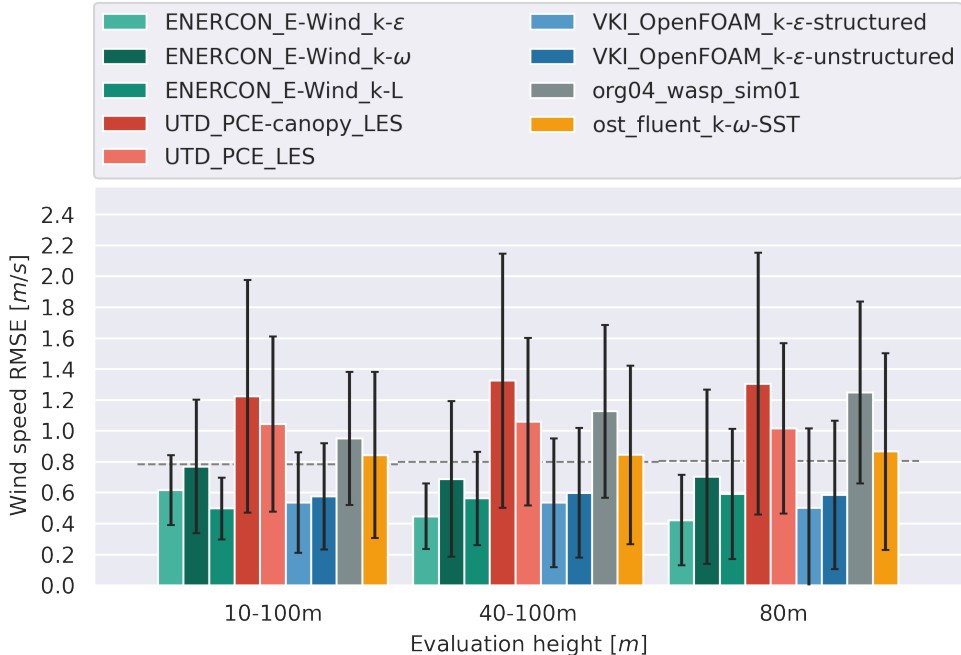

**Figure 5.** Root-mean-square errors for different wind profile evaluation heights for met mast 25. The colored bars denote the average RMSE of the wind profiles in each of the twelve sectors. The vertical black lines denote the standard deviation of the error among the twelve sectors. For evaluation height 80 m the RMSE reduces to the difference between the predicted and measured wind speed.

by around 25 - 40%. An underestimation of almost 35% can be seen for both OpenFOAM simulations in case of met mast 29 and 25, but only 10% in case of met mast 20. The accuracy of WAsP was underestimated by up to 35% in all cases.

Overall, the E-Wind simulations achieved the highest and most consistent skill scores of around 80% for met masts 25 and 20 and only perform slightly worse than the LES and OpenFOAM simulations for met mast 29, which reached skill scores of 85% and 90%, respectively. Taking the cost scores into account, E-Wind is the most suitable tool for the Perdigão site.

One important point to note is that the scores reveal the immense cost needed for LES simulations. Given the lower accuracy compared to simpler models, the additional costs are not justified and so cannot be recommended for wind modellers in the industry. Hence, the application of tools such as LES and DNS will remain solely in the academic field in the foreseeable future. However, LES remains an important tool when dynamic phenomena are relevant, such as in the estimation of wind turbine fatigue life.





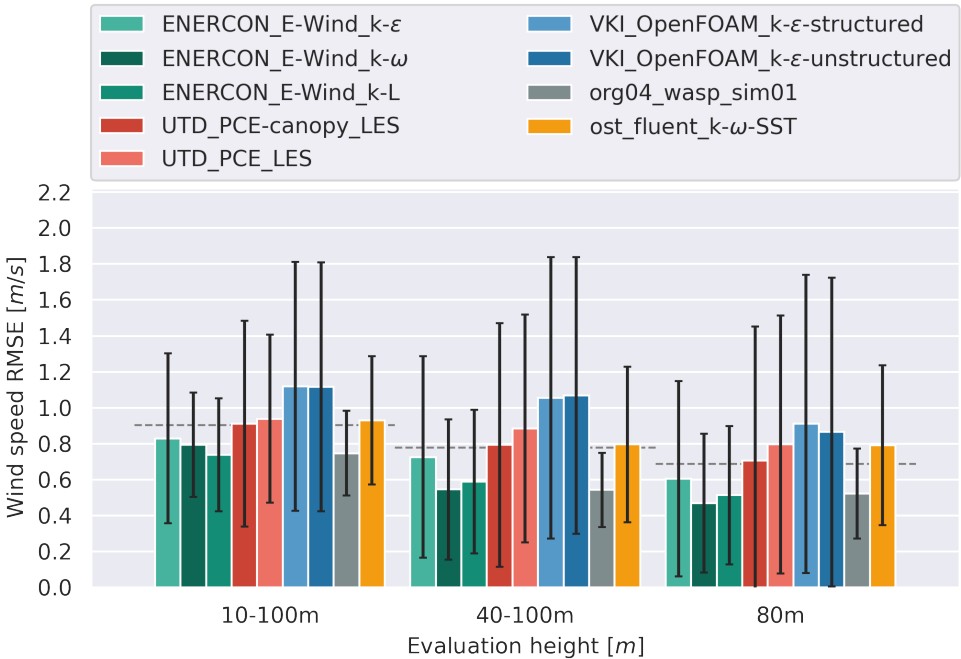

**Figure 6.** Root-mean-square errors for different wind profile evaluation heights for met mast 20. The colored bars denote the average RMSE of the wind profiles in each of the twelve sectors. The vertical black lines denote the standard deviation of the error among the twelve sectors. For evaluation height 80 m the RMSE reduces to the difference between the predicted and measured wind speed.

## 3.2 AEP results

### 3.2.1 Estimated AEP values

Next, we look at the values for the AEP, which are used for the calculation of the skill score, $S_E$. Fig. 8 shows the normalised AEP values for the calibration met mast 29 and the validation met mast 25. For the normalisation the overall expected theoret-

ical gross AEP at each met mast was used. As no power or energy measurement data was available, the AEP was calculated by multiplying the measured time series of ten-minute averaged wind speeds with the wind turbine power curve. This was done for each of the 12 sectors over the whole measurement period of around three months. The resulting values were then additionally scaled to account for one year (Barber et al., 2022a).

The AEP values based on the simulations were then normalised by these calculated AEP values at the respective met mast.

It clearly highlights the relative deviation from the AEP based on the measurement data, where a value of 1.0 indicates an agreement with the measurements.

Notable are the great performances of all simulation workflows for met mast 29, accurately predicting the AEP based on the measurements. The ost_fluent_k-$\omega$-SST simulation shows the best agreement with the measurements, followed by





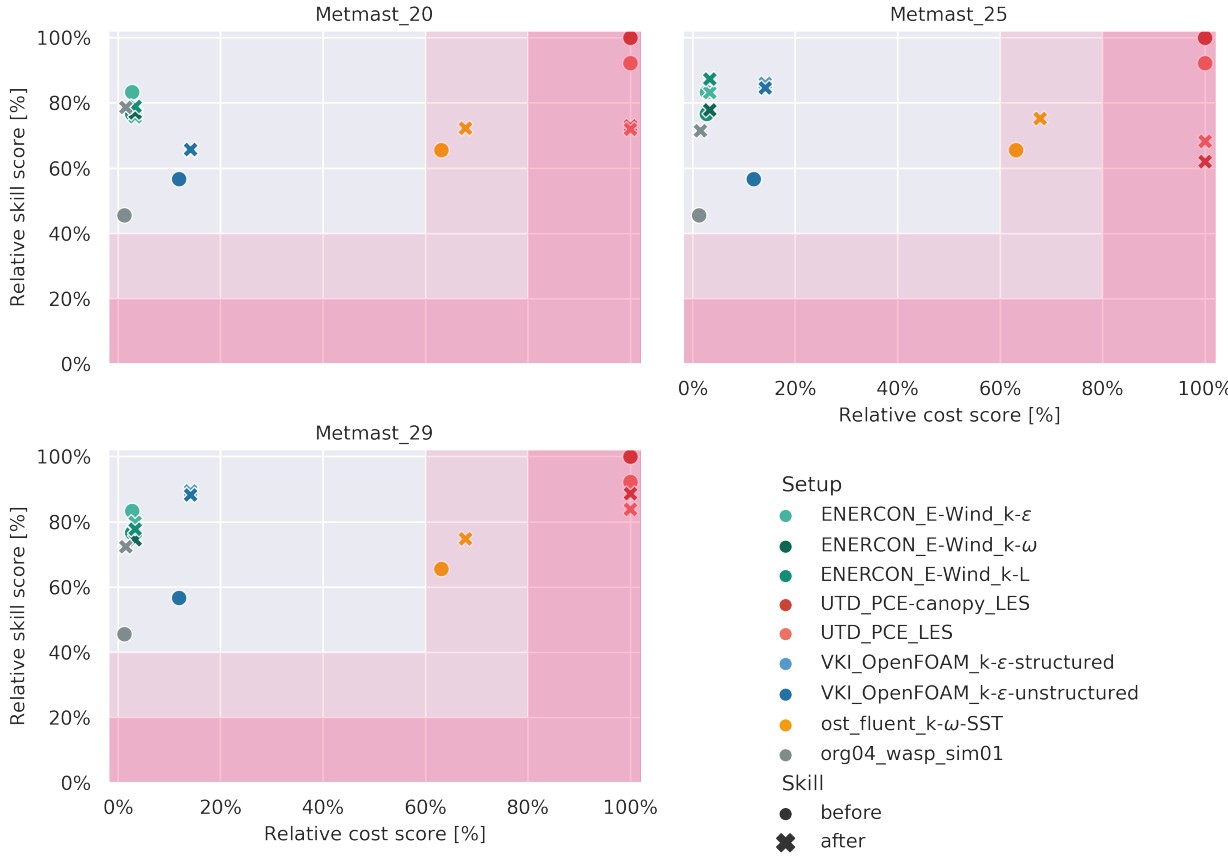

**Figure 7.** The relative skill and cost scores for met masts 20, 25 and 29. The dots denote the "before" scores, estimated based on a questionnaire prior to running the simulations, and the "after" skill scores, calculated based on the wind profile RMSE values for evaluation height 10 - 100 m.

UTD_PCE_LES, without the canopy model. This is in agreement with the low overall wind speed RMSE values for met mast
19, see Fig. 4.

For met mast 25 the E-Wind simulations with the k-L and the k-$\omega$ turbulence models as well as the UTD_PCE_LES without the canopy model show good agreement with the measurements. Interestingly, the performance of the UTD_PCE-canopy_LES is about 40% worse compared to UTD_PCE_LES. A possible reason for this might be the under-prediction of the wind speed at evaluation height 80 m due to the canopy model, which led to a 30% increase in RMSE (Fig. 5). The OpenFOAM simula-
tions VKI_OpenFOAM_k-$\epsilon$-structured and VKI_OpenFOAM_k-$\epsilon$-unstructured show the lowest accuracy and only manage to predict 30% of the overall AEP.

Both simulations use a canopy model, due to which the wind speeds might be heavily under-predicted at met mast 25 and hence lead to the poor performance . These locations are likely to be affected by the forest canopy representation that is



approximated as a uniform porosity throughout the domain. Uncertainties could be reduced by interpolating the forest point
cloud data of Perdigao onto each point of the terrain patch in order to account for heterogeneities in canopy heigh

On the other hand, the WAsP simulation results show a large over-estimation, which might indicate the limits of a linear
model for the chosen complex site.

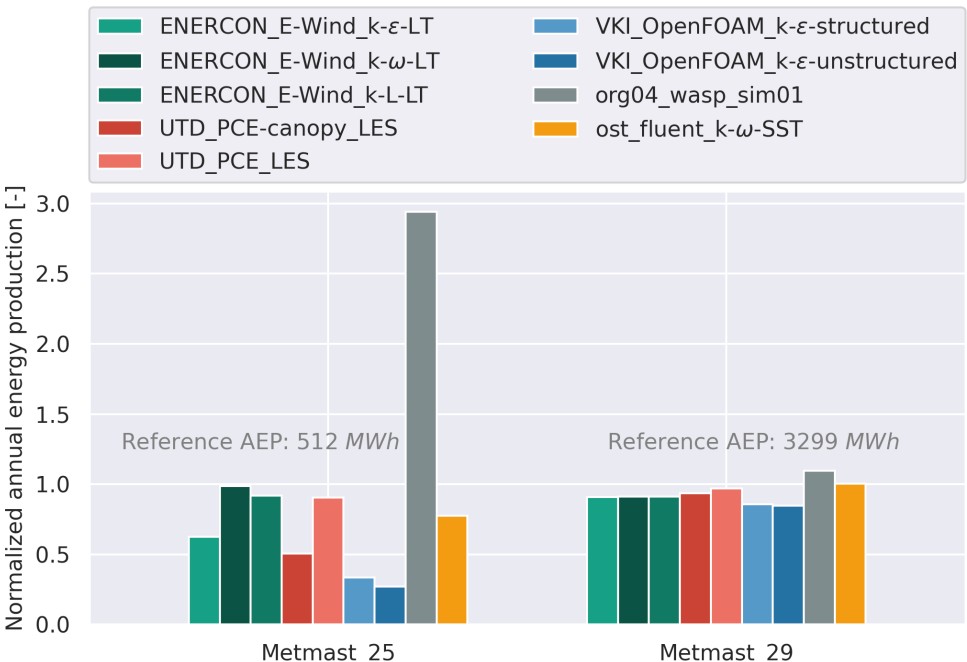

**Figure 8.** Overall predicted AEP values for met masts 25 and 29. A value of 1.0 means that the predicted AEP and the AEP based on the measurement data match. The absolute AEP values for each met mast are given as a reference.

Fig. 9 shows the AEP values for each of the twelve sectors for met mast 29 and met mast 25. For met mast 29 it can be observed that all models somewhat under-predict the energy production for most but the main wind direction, NE. All models
give very similar results for each sector, which is in line with the overall AEP values for met mast 29, see Fig. 8.

For met mast 25, no model is able to accurately predict the AEP for each sector. The main wind direction, SE, is largely under-predicted by all models but UTD_PCE-canopy_LES. The North-Western direction is only accurately captured by E-Wind with the k-L turbulence model. UTD_PCE_LES, UTD_PCE-canopy_LES, ENERCON_E-Wind_k-$\epsilon$ and ENERCON_E-Wind_k-$\omega$ considerably over-estimate the North-Western direction, whereas the remaining models considerably under-estimate
the AEP in the same sector. An under- or over-prediction is especially problematic for sectors with higher wind speeds and wind frequencies, as apparent in the wind roses in Fig. 10. These sectors account for more in the overall AEP estimation and hence should be paid special attention by the modeller. This might also be another explanation for the large differences between the wind speed and AEP estimation errors, as shown in Fig. 12. These discrepancies are currently under further investigation.



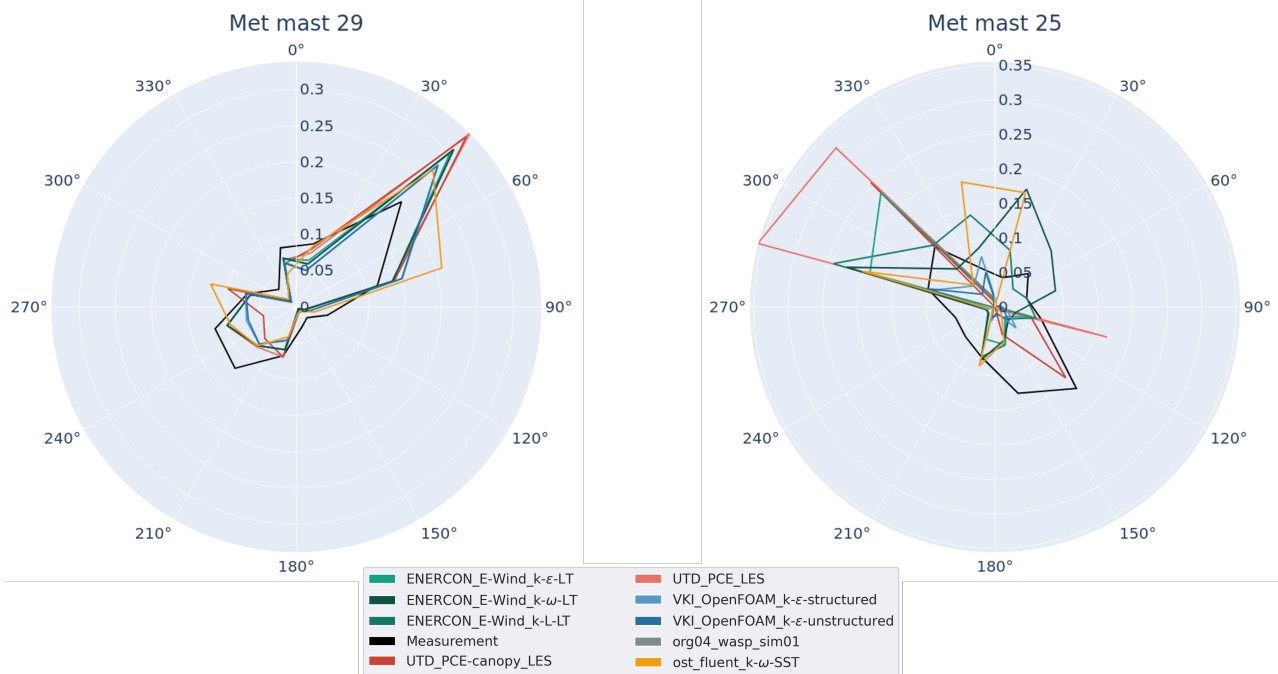

**Figure 9.** Wind roses showing the predicted AEP values for each sector for met mast 29 (left) and met mast 25 (right). The values are normalised by the overall AEP value based on the measurement data of the respective met mast.

### 3.2.2 Skill and cost scores

Fig. 11 show the relative skill versus cost scores based on the AEP for met masts 29 and 25. Similar to the before skill and cost scores for the wind speed simulations, the before skill and cost scores for the AEP estimation are based on a questionnaire filled out by each modeller prior to conducting the simulations. Again, the highest before skill and cost scores are estimated for the LES simulations. For ENERCON_E-Wind_k-$\epsilon$, ENERCON_E-Wind_k-L-LT and ENERCON_E-Wind_k-$\omega$-LT a high skill score of around 80% at a very low cost score of at around 3% are estimated. VKI_OpenFOAM_k-$\epsilon$-structured,

VKI_OpenFOAM_k-$\epsilon$-unstructured and ost_fluent_k-$\omega$-SST have a similar skill score, but differ significantly in cost, mainly due to the high licensing costs for the latter simulation tool. The before skill and cost scores are the same for both met mast positions, as the estimation process is done for the whole site and not for each specific location.

The after skill and cost scores show the actual performances and costs of the simulations. The differences with the respective before scores gives an indication about the soundness of the estimation process based on the questionnaire.

For met mast 29, a very good match between the LES before and after scores can be observed, reaching a skill score of around 95% and a cost score of 100%. The skill score estimation of ost_fluent_k-$\omega$-SST is heavily under-predicted, reaching an after score of 100%, while a before skill score of just 70% was estimated. Given the very high skill score and a cost score of around 65%, this simulation workflow can be deemed better for WRA than the LES simulations, which are considerably more





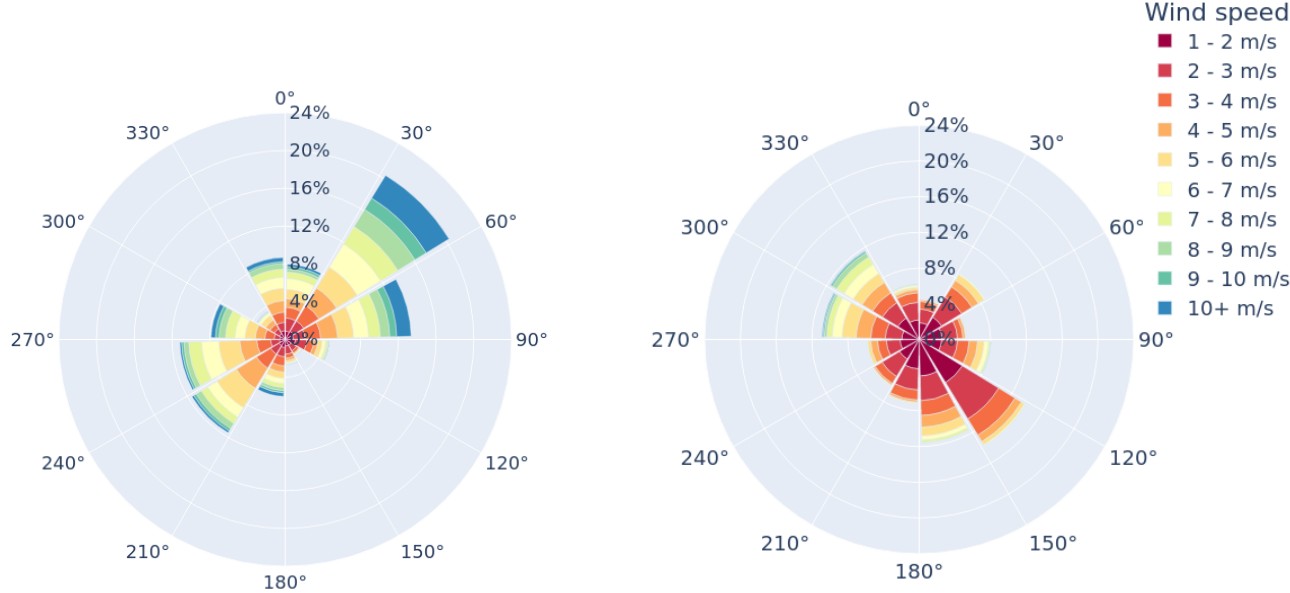

**Figure 10.** Wind roses for met mast 29 (left) and met mast 25 (right) showing the wind speed frequencies for each sector.

costly with a slightly lower accuracy. All three E-Wind simulations (ENERCON_E-Wind_k-$\epsilon$-LT, ENERCON_E-Wind_k-L-
LT and ENERCON_E-Wind_k-$\omega$-LT) as well as the WAsP simulation (org04_wasp_sim01) achieve very high score of around
90% as well, however, with a very low cost score of around only 3%. For the met mast 29 location, WAsP and E-Wind, offer a
great skill score for a comparatively low cost, when considering AEP predictions.

For met mast 25, only the before and after scores for UTD_PCE_LES match very well, with an after skill score of 90% and
a cost score of 100%. The before and after skill scores for ost_fluent_k-$\omega$-SST are much closer to each other for this location,
with an after skill score of around 75% and a cost score of around 65%. Here, the choice between ost_fluent_k-$\omega$-SST and
UTD_PCE_LES is more difficult and it is at the discretion of the modeller to decide whether an increase in cost of 35% can be
justified by an increase in accuracy of 15%. VKI_OpenFOAM_k-$\epsilon$-structured and VKI_OpenFOAM_k-$\epsilon$-unstructured result
in fairly low after skill scores of around 30 - 35%. The highest after skill score of 100% is achieved by ENERCON_E-Wind_k-
$\omega$-LT, followed by ENERCON_E-Wind_k-L-LT, with a skill score of 92%. Interestingly, ENERCON_E-Wind_k-$\epsilon$-LT, with
the k-$\epsilon$ turbulence model, is considerably worse, with a skill score of just 63%. The WAsP simulation, org04_wasp_sim01,
failed to give any reasonable AEP prediction, scoring 0%. The most cost effective and accurate model for the met mast 25
location is ENERCON_E-Wind_k-$\omega$-LT.

Given the great skill and cost scores for both met mast locations, ENERCON_E-Wind_k-$\omega$-LT, can be regarded as the most
promising tool for WRA for the whole Perdigão site.



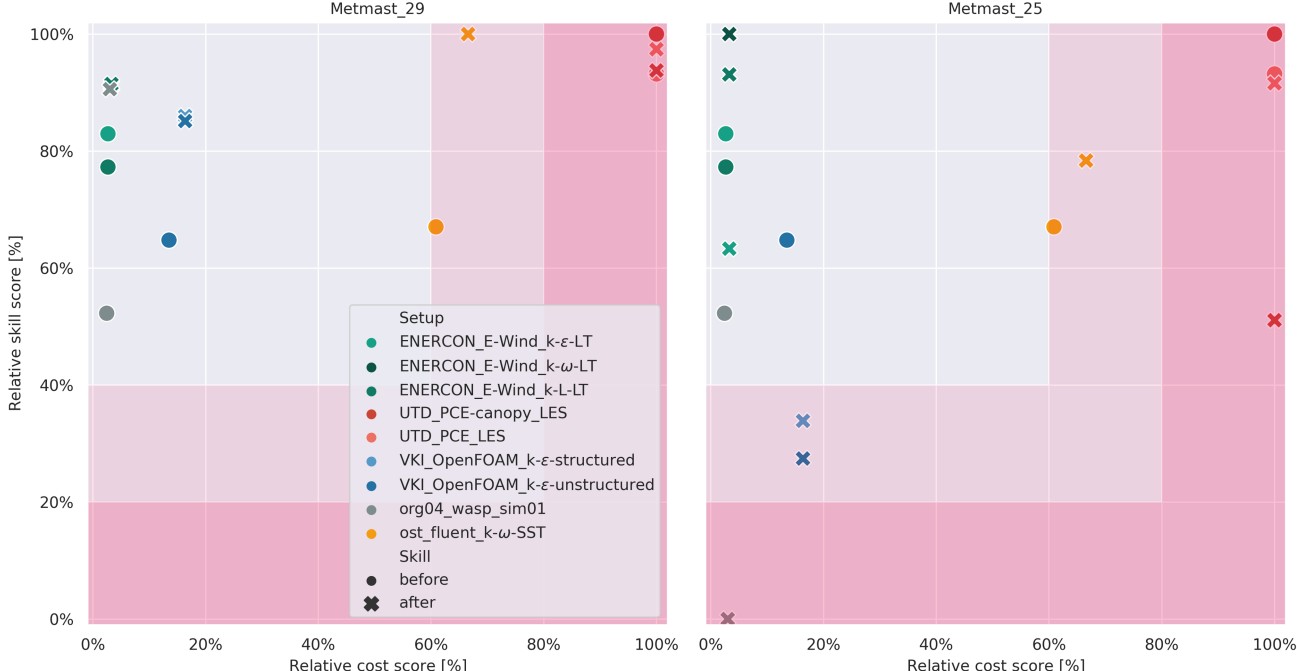

**Figure 11.** The relative skill and cost scores for met masts 20, 25 and 29. The dots denote the "before" scores, estimated based on a questionnaire prior to running the simulations, and the "after" skill scores, calculated based on the predicted AEP values.

## 3.3 Wind speed error vs. AEP error

Another very interesting point is that there does not seem to be an apparent trend between the wind speed RMSE values and the accuracy in AEP predictions, especially when looking at the different simulation tools in Fig. 5 and Fig. 8. For example, the UTD_PCE_LES simulation shows much higher RMSE values, by a factor of two, compared to VKI_OpenFOAM_k-$\epsilon$-structured and VKI_OpenFOAM_k-$\epsilon$-unstructured, at evaluation height 80 m. For the AEP prediction at the same met mast, however, is considerably more accurate, i.e. around 10% error compared to the up to 75% error for VKI_OpenFOAM_k-$\epsilon$-unstructured. This might be due to the different simulation and AEP calculation approaches, as described in Section 2.3.

Fig. 12 shows the comparison between the wind speed prediction error to the AEP prediction error for both met mast 29 (left) and met mast 25 (right). Due to the small sample size and the low correlation values no statistically significant correlation between the two variables could be found for met mast 29, which was also observed by (Barber et al., 2022a). However, for met mast 25 the correlation between the two error measures is much higher, leading to a statistically significant correlation (p-value = 0.026). Nevertheless, this indicates that the wind speed prediction accuracy is not the only variable that influences the AEP prediction accuracy.

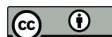

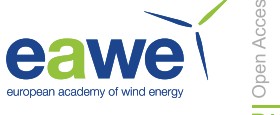
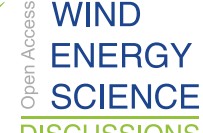

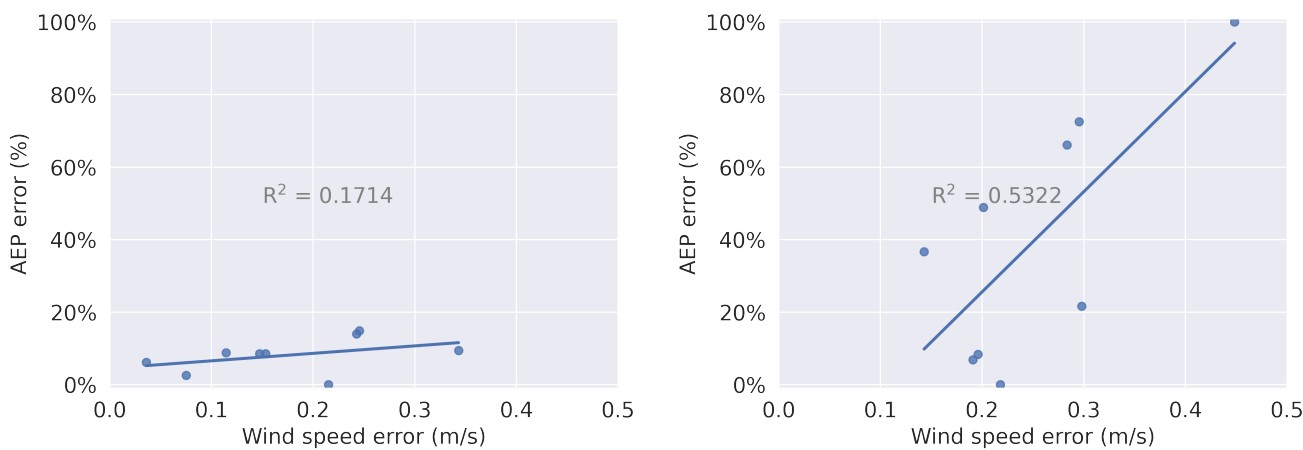

**Figure 12.** Correlations between the wind speed error and the AEP error for all simulation workflows for met mast 29 (left) and met mast 25 (right).



# 4 Conclusions

The goal of this work was to further enhance the understanding and building of a knowledge base for choosing the most optimal WRA tool for a particular complex terrain site based on accuracy and costs. This was done by comparing different simulation tools and WRA workflows for the Perdigão site in Portugal, in terms of accuracy and costs. In total nine different simulations from five different simulation tools were carried out and compared. For the comparsion, the wind profile prediction and the AEP prediction errors were used. These were eventually translated into relative skill and cost scores, allowing the user to set accuracy and cost thresholds for choosing the right tool for the given task. The resulting prediction errors and scores where then compared for all tools and workflows at three met mast positions. It was shown that with a high degree of automatisation, reducing the costs, and a high experience level of the modeller, increasing the skill, a cost effective and accurate prediction based on RANS could be achieved. LES simulations are still mainly reserved for academic purposes due to high computational time and costs, which restrict the number of possible simulations. However, LES remains an important tool when dynamic phenomena are relevant, such as in the estimation of wind turbine fatigue life. This in turn negatively affected especially the AEP prediction accuracy and skill scores. Canopy models have proven to be useful when considering the wind profile prediction accuracy, where profile locations close to the ground are taken into consideration. However, the canopy models used within this study also showed an under-estimation of wind speeds at higher wind speed profile locations, leading to a reduced accuracy for the AEP predictions.

Lastly, for met mast 29 no statistically significant correlation between the wind speed prediction error and the AEP prediction error was found, whereas for met mast 25 a higher wind speed error could at least partly explain an increase in AEP error. However, more simulations are needed in order to increase the sample size and get a better picture of the correlation between these two variables and to identify further contributing variables.





*Data availability.* The data that support the findings of this study are openly available on the Perdigão Field Experiment website at https://perdigao.fe.up.pt/. The datasets used can be found at https://perdigao.fe.up.pt/datasets/thredds/catalog/flux/catalog.

*Author contributions.* Sarah Barber and Florian Hammer managed and coordinated the project. Florian Hammer was responsible for writing the paper. Sebastian Remmler carried out the E-Wind simulations. Federico Bernardoni carried out the PCE-LES simulations. Kartik Venkatraman carried out the OpenFOAM simulations. Alain Schubiger carried out the Fluent simulations. Gustavo A. Diez Sanchez carried out the WAsP simulations. Trond-Ola Hågbo, Sophia Buckingham and Knut Erik Giljarhus assisted Kartik Venkatraman with the simulations, post-processing and analysis.

*Competing interests.* There are no competing interests

*Acknowledgements.* This work was carried out as part of the project "Comparison metrics simulation challenge" funded by the Swiss Federal Office of Energy (SI/501955-01). The E-Wind simulations were kindly supported by the German Federal Ministry for Economic Affairs and Energy within the project "Schall_KoGe".





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
