# Peer review of "Comparison Metrics Microscale Simulation Challenge for Wind Resource Assessment"

_Wind Energy Science, 2022_

## Author Comment (AC1)

**Response to the reviewer**

Dear Rogier Floors,

 We thank you for your useful comments and the time invested in reviewing our manuscript. We have addressed each of your comments as detailed point by point below, which we believe has significantly improved the quality of the manuscript.

**Main Comments**

**Reviewer Point P1** — The paper presents a case study of different models for the Perdigao campaign. It is a noble goal to quantify the accuracy of a model and the resources used to run that model, but unfortunately for me to trust the conclusions provided in the paper I lack important details about both the model setups and the way the 'costs' are calculated. I think the paper should stay clear from drawing too general conclusions about what model is most 'promising' and instead only present the very specific cases for which the models are validated (e.g. from mast to mast). So in it's current form I cannot recommend the paper for publication. I think major revisions are required to rethink the structure and/or allow other researchers to reproduce the results.

**Reply**: Regarding the point "I think the paper should stay clear from drawing too general conclusions about what model is most 'promising' and instead only present the very specific cases for which the models are validated", we have decided to restructure the paper accordingly. Indeed, we agree that the analysis of "costs" and "skill" is very difficult to generalise, and there were many assumptions made in this work. The new goal of the work is therefore now "to build a simulation database that will help to develop and improve WRA decision tools". We have adjusted the introduction and the analysis section accordingly. The analysis now avoids specific recommendations regarding the suitability of each model; rather, it now focuses on the difficulties and learning outcomes for WRA decision systems, which could be used in the future.

**Reviewer Point P2** — I am afraid that with so many models it will be very hard to describe all of the setups, without making the paper 50 pages long. Perhaps a possibility is to put detailed model setups in the appendix. Alternatively, the text should be adapted so that the model setups are provided with enough level of detail to redo the simulations (see detailed comments below).

**Reply**: It is indeed very difficult to find a balance between describing a simulation setup in a detailed enough manner without stretching a paper to 50+ pages. Especially with the amount of different tools and work flows, as presented within this work. We have added more details and/or links to further resources describing the setups and hope this is sufficient to reproduce the simulations.

**Reviewer Point P3** — I also think that In a study like this where no new theory is being presented (which is fine) it is particularly important that the data are openly available so that others can still benefit from the study. So I would suggest expanding the "Data availability." section, with more than just the repository of all Perdigao data.

**Reply**: As there was no further data preparation and filtering done on the Perdigao data itself, we simply referred to the repository, which provides all the necessary data used within this work.

 However, in order to be able to compare simulation results, we made results of all simulations available on ZENODO `https://doi.org/10.5281/zenodo.8005621`.

**Further Comments**

**Reviewer Point P4** — l16: upper heights sounds a bit strange. I suggest higher heights or something similar.

**Reply**: Has been changed accordingly.

**Reviewer Point P5** — l26: long-term wind resource extrapolation: I would call this long-term wind resource correction, the way it is written it seems like the long term wind resource needs extrapolation, but is the shorter term measurements that need to be extrapolated to a longer term climate.

**Reply**: Agreed, wrong term was used here. Has been changed accordingly.

**Reviewer Point P6** — l94: Will this turbine cause any wake effects? This is not discussed.

**Reply**: This has not been considered due to the very low frequency of wind speeds from this direction. Info added to the paper (l. 95).

**Reviewer Point P7** — l149: Are all model outputting time series? Is this generic or which time series are you talking about here?

**Reply**: Here we talk about the measurement data only. This has been made more clear in the text on line 150.

**Reviewer Point P8** — l160: topography -¿ orography (topography is usually defined to include the roughness of the terrain)

**Reply**: Thank you for the clarification. Has been changed accordingly.

**Reviewer Point P9** — Table 1: Common application range: I would rather call this complex or non-complex instead of flat or non-flat. The linearized model will probably work fine in non-flat terrain as long as no flow separation occurs.

**Reply**: As the term "complex terrain" can be interpreted very differently depending on the used simulation tool, we tried to refrain from using it, see https://doi.org/10.5194/wes-7-2231-2022 Instead we mentioned flow separation as a limiting factor. This has been added to line 179.

**Reviewer Point P10** — l177: Who is 'our' here?

**Reply**: Has been changed to "UTD in-house code"

**Reviewer Point P11** — l187: terrain topography - orography

**Reply**: Has been corrected.

**Reviewer Point P12** — l190: Corine Land Cover (European Union, 2018) database: Did you use raster or vector data? Which projection was used? Which datum?

**Reply**: Vector data of the Corine Land Cover (CLC) 2018, Version 2020_20u1 (`https://land.copernicus.eu/pan-european/corine-land-cover/clc2018?tab=metadata`) was used to obtain vegetation data about the area. See line 214.

**Reviewer Point P13** — l190: "The stem size and distribution reproduce the same canopy frontal solidity (Monti et al., 2019; Nepf, 2012) of the actual vegetation at the site extracted from the Corine Land Cover (European Union, 2018) database". This is not clear: what is the actual vegetation at the site? How can you get that from CORINE data which is just a satellite based product? How can that match the stem size and distribution?

**Reply**: Additional information has been added to the section.

**Reviewer Point P14** — PCE-LES: which source did you use for the terrain elevation?

**Reply**: Additional information has been added to the section.

**Reviewer Point P15** — Section 2.3: after reading this I was expecting all modellers use the same terrain elevation, but based on l198 I start to doubt that because there SRTM is mentioned. In the LES section no source is mentioned.

**Reply**: The choice of how and what to use for modelling the terrain elevation was left to the modeller. As this is highly dependent on the used tools. This has been clarified in the text, see line 172.

**Reviewer Point P16** — l215: Scaled how? You mean assuming the wind distribution for the three months is representative for the whole year?

**Reply**: This is correct. A remark has been added for clarification.

**Reviewer Point P17** — l218: That reference does not really describe the WAsP stability model. Better to cite Troen and Petersen (1989). What stability setting are used in the end?

**Reply**: Thank you for the reference. This has been changed. WAsP wind modelling variables were set by default according to the atmospheric stability effects obtained.

**Reviewer Point P18** — l219: What was the source of this tiff data?

**Reply**: Information has been added to the text.

**Reviewer Point P19** — l220: What is in the .map file? What was the source?

**Reply**: Information has been added to the text.

**Reviewer Point P20** — l221: There is several roughness tables in that reference? Which one was used?

**Reply**: Information has been added to the text.

**Reviewer Point P21**  — l223: The direction variable: you mean wind direction? From which height? What is NCA?

**Reply**: Sentence has been changed for clarification. Reference was fixed as well. Meant was the Earth Observing Laboratory.

**Reviewer Point P22**  — l225: Which 'data'?

**Reply**: Information has been added to the text.

**Reviewer Point P23**  — l225: References for MERRA and ERA5 missing

**Reply**: Information has been added to the text.

**Reviewer Point P24**  — l226: Coefficient of determination between what and what?

**Reply**: Information has been added to the text.

**Reviewer Point P25**  — l226: How do you define "very similar"?

**Reply**: Information has been added to the text.

**Reviewer Point P26**  — l226: "some basic filtering", "constant line values", "some variables": specify what filtering, what is a constant line value, which variables?

**Reply**: This has been made more clear in the text.

**Reviewer Point P27**  — l229: Section on stability: this can also be made quantitative.

**Reply**: Information has been added to the text.

**Reviewer Point P28**  — l229: What kind of adjustment?

**Reply**: Information has been added to the text.

**Reviewer Point P29**  — l230: What was optimized with respect to what?

**Reply**: Information has been added to the text.

**Reviewer Point P30**  — l232: Please specify in more detail what kind of long-term correction you did.

**Reply**: Information has been added to the text.

**Reviewer Point P31**  — l246: what was the upper domain boundary?

**Reply**: We specified the height of the E-Wind domain (6 km) in the manuscript. Furthermore, we added some details on the used boundary conditions.

**Reviewer Point P32** — l246: Is the first bin centered around north or from 0 to 15 degrees?

**Reply**: In the RANS simulation, we do not consider sectors, but discrete wind directions. The first geostrophic wind direction is 0°. We added this information to the manuscript.

**Reviewer Point P33** — l247: How do you define the wind shear?

**Reply**: The definition was added to the manuscript.

**Reviewer Point P34** — l255: Similar as my previous comment: so then you assume the 3 month wind measurements are representative of the full year? That is fine, but it is inconsistent with the previous model setup (Windpro), where you apply a long term correction.

**Reply**: For the wind speeds no long term corrections were performed. Only the energy production values that were calculated for three months was scaled to obtain the annual energy production.

**Reviewer Point P35** — l261: roughness height -¿ roughness length

**Reply**: Thanks for spotting. Has been corrected.

**Reviewer Point P36** — l265: Is the roughness length varying with wind direction sector?

**Reply**: This has been made more clear by adding "for all wind directions".

**Reviewer Point P37** — l267: grid independence study -¿ I assume the conclusion of the grid independence study was that the simulations were not dependent on resolution? Why was the resolution of 15 m optimal? In which sense was it optimal?

**Reply**: The sentence "For resolutions below 15 m no change in wind speed profiles compared to the 15 m resolution was observed." has been added

**Reviewer Point P38** — l285-l289: Taking the mean of a RMSE is mixing different errors metrics. You should calculate the squared errors from each sector and do the root-mean in the last step?

**Reply**: The sentence has been changed for clarification. What is meant here is that the RMSE value for each simulation was taken and then averaged to make a relative comparison between the simulations. The calculated RMSE value for each simulation, however, was calculated as you indicated.

**Reviewer Point P39** — l292: m/s should be in normal font

**Reply**: Thanks for spotting. Has been corrected.

**Reviewer Point P40** — l302: it would be good to mention here that this is the calibration point.

**Reply**: Sentence has been added.

**Reviewer Point P41** — l318: This is not that surprising because both masts are located on top of a hill. It would be useful to relate this to the "most similar predictor" discussion in https://wes.copernicus.org/articles/5/1679/2020/.

**Reply**: Thank you very much for this reference. However, as the performance differences, i.e. either better or worse, between met mast 25 and met mast 20 vary quite a lot for the various simulations we would refrain from relating this to the "most similar predictor".

**Reviewer Point P42** — Fig 4: I am bit confused how big the errors are at 80 m. Wasn't that used for calibration? How can there be already RMSE of up to 0.4 m/s?

**Reply**: For the OpenFOAM and the Fluent simulations the calibration was done at a height of 100 m. E-Wind did the calibration based on wind shear, whereas for the LES simulations only 8 wind directions were simulated and then interpolated to obtain the desired results for 12 sectors. WAsP performs internal corrections, especially in complex terrain, which can slightly vary the results. These information, where missing or wrong, have been added to the respective model descriptions.

**Reviewer Point P43** — Sect 3.1.2: This section is hard to understand ; what is the main message? The relative costs and skill score appear suddenly in Fig 7, but it feels like some background on the numbers should be available (appendix?). As discussed the costs and skill are extremely hard to quantify, so you could end up with any ranking of the models here. I would avoid drawing conclusions like "Taking the cost scores into account, E-Wind is the most suitable tool for the Perdigão site.".

**Reply**: The methodology for this was developed and presented in Barber et al. (2022b), whereas we applied this methodology to our case study and simply presented the results. Based on and in the context of this methodology the terms 'cost', 'skill', 'most promising', 'most suitable tool', etc. was used. Having said that, in the processing of restructuring the paper and changing the goal, as mentioned above, this section has now been removed.

**Reviewer Point P44** — l370: mast 19? You mean 29?

**Reply**: Thanks for spotting. Has been corrected.

**Reviewer Point P45** — l380: end of line: height.

**Reply**: Thanks for spotting. Has been corrected.

**Reviewer Point P46** — Fig 8: Are we comparing AEP at the same heights here? That should be added somewhere.

**Reply**: The AEP was calculated and compared at height 80 m. In the section "Skill and cost scores" we changed and clarified some text about the AEP calculation, now also stating that it is determined at a height of 80 m.

**Reviewer Point P47** — l386: What is an AEP by sector? I only know about an AEP as the production for a year, i.e. for all sectors combined.

**Reply**: This is simply supposed to show how important a simulated wind direction is in terms of the contribution to the overall AEP. For some wind directions the wind profiles and the AEP values were underestimated and for some others overestimated, which in some cases can cancel each other out. So

by just looking at the overall AEP one might get the impression that the prediction of the overall AEP is fine, however, is in fact a result of a combination of under- and overestimations for different wind directions. A sentence has been added to make the purpose of the plot more clear, see line 431.

**Reviewer Point P48** — l423: It is for me again not quite clear how this is quantified. I would leave out generalizations like this and just discuss model differences. How does a single AEP prediction from one mast to the other make this the model the best for entire Perdigao site?

**Reply**: We agree that this statement is too bold and quite an extrapolation. Sentence has been deleted.

**Reviewer Point P49** — Fig 11: Where is mast 20?

**Reply**: Unfortunately, the simulation results for the AEP at met mast 20 are not available. This is due to the original design of this case study, with the original goal to only compare met masts 25 and 29 for the AEP predictions. This information has been addded to the text, see line 394.

**Reviewer Point P50** — Sect 3.3: I agree there is so many differences in the difference model chains to calculate AEP that it is impossible to say what it is the exact reason. If mast 29 is used for calibration one would not expect any model error in AEP? So I would just leave this section out.

**Reply**: This section has been reformulated to be more clear and precise. Also see the next point for further details.

**Reviewer Point P51** — l454: It would be a very surprising conclusions if the AEP error did not depend on wind speed error. What about air density? How has that been calculated in the different model chains?

**Reply**: The statement of this section is that the overall wind speed prediction accuracy is not the only variable that influences the overall AEP prediction accuracy, but that looking at sector values, especially for the most frequent wind direction, is important. This is also the reason for introducing the "AEP by sector" plots and how they contribute to the overall AEP.

---

## Author Comment (AC2)

**Response to the reviewer**

We thank the reviewer #2 for their useful comments and the time invested in reviewing our manuscript. We have addressed each of the referee comments as detailed point by point below, which we believe has significantly improved the quality of the manuscript.

**Main Comments**

**Reviewer Point P1** — The topic addressed by the authors is very relevant to the wind energy industry and their "skill/cost" idea is certainly interesting. However, there is a lot of missing information about the organization and structure of the benchmark, making it difficult to assess if the model's comparison and its findings are robust. There are also some caveats described below about the scoring process definitions that prevent me to recommend the publication of this work without some major revisions that can provide more confidence in the conclusions drawn in the paper in its current form.

**Reply**: Regarding the point "There are also some caveats described below about the scoring process definitions", we have decided to restructure the paper accordingly. The new goal of the work is now "to build a simulation database that will help to develop and improve WRA decision tools". We have adjusted the introduction and the analysis section accordingly and the paper now focuses on the difficulties and learning outcomes for WRA decision systems, which could be used in the future. Furthermore, additional information about the "organization and structure of the benchmark" have been added to the article.

**Reviewer Point P2** — First, I believe that the definition of the skill scores given "before" the simulations are not very clearly described in the manuscript. The reader is pointed to a previous work by the authors (Barber et al., 2022b), which was concluded with several suggested improvements and pending work related to the scoring conceptualization. However, those points don't seem to be implemented for the analysis of the present study, or at least this manuscript doesn't explain how the subjectivity in many of the scoring definitions can be mitigated.

Unlike the skills scores, some components of the cost scores are indeed more "quantifiable" (equation 1). Still, the values of some of them like the "cost of the staff training per project" can again be very subjective to the interpretation of the modeler. Other parameters such as the "hourly rate of the modeler" depend on the institution, country, etc. The cost scores assigned in this comparison can be biased towards the participants from countries with lower wages.

**Reply**: We have improved the description of skill scores "before" in Section 2.2. We haven't made the improvements suggested in the previous paper because the work was actually carried out in parallel. However, we have now restructured the paper with a goal of "to build a simulation database that will help to develop and improve WRA decision tools", rather than to present a fully-functioning decision-making method. This has allowed the difficulties with the method to be brought up in the analysis (including the ones you mention here as well as ones brought up by the other reviewer). We have adjusted the introduction and the analysis section accordingly. The analysis now avoids specific recommendations regarding the suitability of each model; rather, it now focuses on the difficulties and learning outcomes for WRA decision systems, which could be used in the future.

**Reviewer Point P3** — On the other hand, besides the challenge of finding an optimal skill/cost model, equally relevant for the wind resource assessment community is the correct usage of the selected model, especially those models very sensitive to the user's expertise such as some research codes included in this comparison. In that regard, I find many technical details missing about the models' setup (see the specific comments), and for those details that are included, it is noticeable the important differences in their configurations beyond their different physics/numerics. For instance, one model includes the effects of the turbine wake (PCE-LES) while another one employed a long-term correction from ERA5 as input wind field (WAsP). Others included atmospheric stability (E-Wind) while others used time series instead of period averages (E-Wind, Fluent). While this is fine in benchmarking exercises such as the CREYAP series (Mortensen et al., 2015), it complicates the goal of the authors of finding an optimal WRA tool because it is very difficult to conclude if the differences in the skill (and costs) among the workflows are related to the model, its configuration or simply the methodology to compute the APE beyond simulating more or less directional sectors. Besides, this mix of factors would probably preclude extrapolating the optimal workflow found in this case study to other sites.

**Reply**: We agree that the correct usage of the selected model is highly relevant to this project, which is why "user skill" is included in the skill score. We have improved the description of this point in Section 2.2 and included a discussion in the analysis in Section 3.1.2.

Regarding the technical details of the setups, we completely agree with your point. However, in order to assess this correctly, we would have to carry out parameter studies for thousands, or millions, of combinations of different parameters, which was beyond the scope of this study. Instead, the idea was to let the users apply their models however they want, in order to demonstrate and highlight how these differences could impact the results. We have added a discussion about this in Section 2.3, see line 172. As well as this, We have added more details, links to further resources and citations describing the setups and hope this is sufficient to reproduce the simulations.

As stated above, the goal of this work was to apply and provide further results of the methodology developed in Barber et al. (2022b) by presenting the obtained results. These results can hopefully help improve the methodology in upcoming work.

**Reviewer Point P4** — The description of the case study is also a bit superficial in section 2.1. In addition to the general description of the site and the measurement campaign, I would suggest that the authors could add more information about the met mast data preparation, filtering and selection. Useful information could be the data availability by direction from the three met masts and comments about the potential effects from the wake of the operating WTG on the mast measurements. The wind rose shown in Fig. 2. indicates less frequent but still important SW winds, for which the wake of the WTG is expected to occur.

**Reply**: As there is an extensive paper by Fernando et al. (2019), describing the measurement data and the flow phenomena of the site we restricted the description within this paper to the most important points. Furthermore, the used met mast data is referenced in the data availability section. No further data preparation or filtering was carried out. However, some additional information regarding wake effects and data availability have been added to the paper, see line 95.

**Reviewer Point P5** — It would be also great if the authors could add information about the atmospheric stability during the 3-month period considered for the benchmark. This information

might help to explain some of the large errors obtained by some of the models. It is already interesting to see that mast 29 has a large deviation from the (neutral stability) logarithmic profile, whereas mast 25, located on the lee side of the hill, thus, with a more complex flow, has an excellent logarithmic fit.

**Reply**:
    We agree that the thermal stability plays an important role in flow over non-flat terrain. A few paragraphs addressing this have been added to the text in Section 2.1.

**Further Comments**

**Reviewer Point P6** — line 56-60: The authors are mentioning the analysis carried out by Barber et al., 2020c. but this work is only available as WECD since that paper has been withdrawn.

**Reply**: Reference has been changed to a published paper.

**Reviewer Point P7** — line 72: By previous works are the authors referring to the Barber et al., 2022a-b articles instead of the Barber et al., 2020a?? Only the first ones compare several modeling tools on different sites.

**Reply**: Yes, the Barber et al., 2022a-b articles are referred to. An additional citation has been added to clarify.

**Reviewer Point P8** — line 81: This citation also points the reader to the withdrawn paper of Barber et al., 2020c. Wouldn't it be "Barber et al., 2022b" the right citation in this case?

**Reply**: Yes, the Barber et al., 2022a-b articles are referred to. An additional citation has been added to clarify.

**Reviewer Point P9** — As mentioned above, this section lacks many details about the model's technical setup. In the case of the RANS and LES-based models, information about their boundary conditions, especially the treatment of the ground, is critical for understanding and potentially allowing the repeatability of this work. Also very important is providing the values of the different models' constants used by the RANS models.

**Reply**: Where possible, the citation of work with further details are provided and more information was added for the different models. Furthermore, the simulation results themselves were made available on Zenodo `https://doi.org/10.5281/zenodo.8005621`.

**Reviewer Point P10** — -the OpenFOAM model is set to match the wind speed and direction at the calibration mast at 100m, (line 212). Aren't all the other models set to match the wind at 80m (line 117)?

**Reply**: Thank you for noticing. The sentence stating that the models were set to match the wind speed at 80m was misleading and has been removed. More details about the exact calibration method for each model is given in the respective model section. The calibration process is highly dependent on the used tool.

**Reviewer Point P11** — Line 223. Change "The direction variable" for "the wind direction". And, where is this wind direction obtained from?

**Reply**: The wind direction was obtained based on the given met mast data, providing the 'U' and 'V' wind speed components. This has been made clearer in the text.

**Reviewer Point P12** — Line 223. Change "Wind speed variables" for "wind speed components"

**Reply**: Has been corrected.

**Reviewer Point P13** — Line 227. I think that the phrase "constant line values for some variables" is not clear about which variables are they referring to, and what "constant line" means in this context.

**Reply**: Sentence has been changed to be more clear.

**Reviewer Point P14** — Line 246. Does the E-Wind workflow simulate 24 wind directions? or 12 as described in Table 1?

**Reply**: Thanks for noticing. 24 wind directions were simulated and Table 1 has been corrected.

**Reviewer Point P15** — Figure 4. Despite that is it expected to have some small errors in the RANS and LES models, shouldn't the WAsP model have no error at the calibration mast at 80m due to the way this model works??

**Reply**: No errors at the calibration mast are very difficult to obtain, because we are comparing simulated data with measured data, although this is at the same point. WAsP performs internal corrections, especially in complex terrain, which can slightly vary the results, as in this case.

**Reviewer Point P16** — Line 364. Is "respective met mast" the calibration mast?? This phrase is a bit confusing, so, it is not very clear how this normalization is done.

**Reply**: This was done for each met mast, i.e. 20, 25 and 29. This has been clarified in the text.

**Reviewer Point P17** — Figure 8. Is this the AEP at 80m height? Again, I'm not sure if I got how the AEP normalization was defined.

**Reply**: Correct, this is the AEP at height 80m. The AEP was normalised by diving the total AEP of the simulations at a met mast by the calculated AEP based on the measurements at that met mast. The clarification mentioned in the previews point hopefully helps to understand this better.

**Reviewer Point P18** — Figure 11. It seems that the mast 20 is missing.

**Reply**: Unfortunately, the simulation results for the AEP at met mast 20 are not available. This is due to the original design of this case study, with the original goal to only compare met masts 25 and 29 for the AEP predictions. This information has been addded to the text, see line 394.